# Estimation of Functional Fitness of Korean Older Adults Using Machine Learning Techniques: The National Fitness Award 2015–2019

**DOI:** 10.3390/ijerph19159754

**Published:** 2022-08-08

**Authors:** Sang-Hun Lee, Seung-Hun Lee, Sung-Woo Kim, Hun-Young Park, Kiwon Lim, Hoeryong Jung

**Affiliations:** 1Division of Mechanical and Aerospace Engineering, Konkuk University, 120 Neungdong-ro, Gwangjin-gu, Seoul 05029, Korea; 2Physical Activity and Performance Institute, Konkuk University, 120 Neungdong-ro, Gwangjin-gu, Seoul 05029, Korea; 3Department of Sports Medicine and Science, Graduate School, Konkuk University, 120 Neungdong-ro, Gwangjin-gu, Seoul 05029, Korea; 4Department of Physical Education, Konkuk University, 120 Neungdong-ro, Gwangjin-gu, Seoul 05029, Korea

**Keywords:** smart fitness, support vector regression, random forest, XGBoost, artificial neural network

## Abstract

Measuring functional fitness (FF) to track the decline in physical abilities is important in order to maintain a healthy life in old age. This paper aims to develop an estimation model of FF variables, which represents strength, flexibility, and aerobic endurance, using easy-to-measure physical parameters for Korean older adults aged over 65 years old. The estimation models were developed using various machine learning techniques and were trained with the National Fitness Award datasets from 2015 to 2019 compiled by the Korea Sports Promotion Foundation. The machine-learning-based nonlinear regression models were employed to improve the performance of the previous linear regression models. To derive the optimal estimation model that showed the best estimation accuracy, we developed five different machine-learning-based estimation models and compares the estimation accuracy not only among the machine learning models, but also with the previous linear regression model. The coefficient of determination of the FF variables was used to compare the performance of each model; the mean absolute percentage error (MAPE) and standard error of estimation (SEE) were used to evaluate the model performance. The deep neural network (DNN) model presented the best performance among the regression models for the estimation of all of the FF variables. The coefficient of determination in the HGS test was 0.784, while those of the others were less than 0.5 meaning that the HGS of older adults can be reliably estimated using easy-to-measure independent variables.

## 1. Introduction

The number of elderly people older than 65 years of age has increased rapidly in recent decades, and has been a critical social issue in many countries [1]. Human aging leads to the degradation of physical functionality, such as weakening of the muscle forces, and can cause serious diseases and impairments [2,3]. The degradation of physical functionality is critically correlated with diseases that arise in old age. For example, the timed up-and-go (TUG) test results of elderly people show a correlation with their mental and physical health [4]. Health-related quality of life is associated with physical functionality, and it is recommended that physical functionality is maintained in old age in order to ensure a healthy life without illness [5].

Habitual physical activity and proper nutrition are the top priorities for delaying the loss of physical functionality [6]. The World Health Organization (WHO) recommends that elderly individuals perform moderate-intensity aerobic exercises for at least 150–300 min per week, vigorous-intensity aerobic exercise for at least 75–150 min, or a combination of both. The WHO also recommends strength training for more than 2 days a week and multicomponent physical activities more than 3 days a week [7]. Although regular physical activity is important to preserve a healthy life in old age, <40% of the elderly population exercise regularly in their daily life [8]. Insufficient physical activity primarily causes functional disability and limited mobility, and these cascades result in more critical illnesses [9]. Smart fitness services that provide personalized workout programs can be an effective solution to encourage the elderly population to exercise regularly. A personalized workout program that fits an individual’s physical ability prevents over- or under-exercise and cramping during exercise. Estimation of physical fitness levels, including the muscle strength, flexibility, coordination, agility/dynamic balance, and aerobic endurance of elderly persons, is important to construct personalized workout programs, and functional fitness (FF) assessment tests have been used to evaluate individual physical fitness levels [10].

However, performing the FF assessment and monitoring the physical abilities periodically in older adults are associated with many difficulties. In addition, measuring FF variables requires a sophisticated device and is costly. To address these inconveniences, several previous studies have proposed methods for estimating an individual’s FF variables with simple physical parameters using multiple linear regression (MLR). Nevertheless, the MLR model has a critical limitation in that it can only represent the linear relationship between the inputs and outputs. In previous studies, machine-learning-based prediction models, such as support vector machine (SVM) and random forest (RF), have been used to consider nonlinear relations in FF prediction. Mahajan et al. reported that the RF model improved the estimation accuracy of 231 divers’ physical fitness levels compared with the linear regression model [11]. Akay et al. predicted the hamstring and quadriceps strength of athletes using an SVM estimation model [12]. Zhu et al. used SVM to predict athletes’ performance [13], Taha et al. estimated archers’ physical fitness level using the k-nearest neighbors and SVM [14], and Matteo et al. proposed nearest neighbor models to predict athlete performance in team sports [15].

Nevertheless, these nonlinear prediction models focused on being trained with individuals who have superior physical functionality, which is not the general population. Previous studies have measured athletes’ physical information to train a prediction model with complicated equipment, which is not adequate for ordinary applications. Lee et al. presented an artificial neural network-based regression model for Korean adults aged <65 years [16]. However, they did not consider older adults whose variables in the FF test were different. In this paper, we propose a machine-learning-based estimation model for FF variables with easy-to-measure physical variables in Korean older adults. To derive the optimal estimation model that shows the best estimation performance, we constructed various machine-learning-based estimation models and evaluated the estimation performance of each model.

The main contributions of this study are as follows: This study proposed the FF variable estimation model for evaluating the physical fitness level of elderly adults using easy-to-measure independent variables. The proposed model can be used as an effective tool to evaluate the personal fitness level in smart fitness services.Various nonlinear machine learning regression models were constructed and evaluated to compare the accuracy with the previous linear model and to derive the optimal estimation model presenting the best estimation performance.

## 2. Materials and Methods

### 2.1. Ethics Statement

The study was conducted in accordance with the guidelines of the Declaration of Helsinki and was approved by the International Review Board of Konkuk University (7001355-202101-E-132).

### 2.2. Dataset

The National Fitness Award (NFA) is a program carried out by the Ministry of Culture, Sports, and Tourism (MCST) and the Korea Sports Promotion Foundation (KSPF) to measure the physical fitness levels of general Koreans aiming to help people live healthier. This paper employs the NFA datasets of elderly adults (age: ≥65 years), gathered by the KSPF to train the machine-learning-based estimation models. The NFA dataset includes the physical fitness levels of individuals that are measured under strict measurement protocols at 75 sites throughout the Republic of Korea. The participants of the NFA dataset, who were collected between 2015 and 2019, were older adults (total = 210,490) who were older than 65 years of age. We excluded missing values of older adults’ independent variables and FF variables, resulting in 178,960 adults in the regression model datasets (men: *n* = 61,465, women: *n* = 117,495). The regression models in the study used independent variables (e.g., sex, age, body mass index (BMI), and percent body fat) as the inputs and predicted the FF variables, including hand grip strength (HGS), lower body strength (30 s chair stand test), lower body flexibility (chair sit-and-reach test), coordination (figure-of-eight walk test), agility/dynamic balance (TUG test), and aerobic endurance (2 min step test), as the outputs; 70% of the data (total: *n* = 125,272, men: *n* = 42,911, women: *n* = 82,281) were used as the training dataset, and 30% of the data (total: *n* = 53,688; men: *n* = 18,474; women: *n* = 35,214) were used as the validation dataset. A summary of the NFA datasets is presented in Table 1.

Measurement of physical independent variables and FF variables: The measurements of independent variables and FF variables followed the NFA guidelines, as presented in a previous study [17].

### 2.3. Data Pre-Processing

Pearson’s correlation analysis was used to assess the linear relationship between the independent variables and FF variables. Table 2 shows the degree of correlation between the input and output variables. HGS had a positive linear correlation with height and weight and a negative correlation with sex and percent body fat. Figure-of-eight walk and TUG had a positive correlation with age, and the 30 s chair stand had a negative correlation with age. The chair sit-and-reach showed a positive correlation with sex and a negative correlation with height. As most of the FF variables showed a weak linear relation with the independent variables, it is possible to consider using nonlinear prediction models rather than linear prediction models to improve the prediction accuracy. 

Standardization: Standardization, which is a feature scaling technique, was used for the input variables to avoid data redundancy and dependency caused by feature scale differences (Equation (1)). All data, except sex, were centered around the mean of 0 with a standard deviation of 1.
(1)$${\hat{x}}_{i}=\frac{x_{i}-\mu }{\sigma }$$
where $x_{i}\;\mathrm{and}$ ${\hat{x}}_{i}$ denote the values of the input and standardized input, respectively. μ and σ are the average and standard deviation of the input variable $x_{i}$, respectively. For standardization, sex was expressed as 1 (male) or 2 (female).

Outlier removal: Outliers, which can distort statistical analyses and create prediction models of poor outcomes, are data with abnormal values from other data. To manage outliers in the training dataset, the studentized residual (SRE) was used, and outlier data were removed when the absolute value of the SRE was >2 [17].

Feature selection: Feature selection methods were used to increase the estimation performance and shorten the training regression model. The *p*-value was used to validate the relationship between the independent variables and FF variables. The independent variables with *p*-value is >0.05 were removed for dimension reduction and estimation accuracy improvement. In addition, feature selection using the Boruta algorithm was used to assess variables that could decrease the performance of the regression model and cause overfitting. In the ranking of the features, 1 means confirmed, 2 means tentative, and 3 means rejected. The *p*-values for each variable and the Boruta algorithm ranking are listed in Table 3. With reference to feature selection, we selected the input variables and maximized the estimation performance.

### 2.4. Machine Learning-Based Estimation Models

Various machine-learning-based regression models were used to predict the FF variable with independent variables. Each model was evaluated using R^2^ and SEE values and was compared with the other models. A summary of this method is shown in Figure 1.

#### 2.4.1. Support Vector Regression

SVM, which predicts the optimal hyperplane generated in an n-dimensional feature space, is a supervised learning algorithm for classification and regression. SVR is specifically used for regression, and Equation (2) represents the linear approximation function [18].
(2)y=ω·x+b
where ω is the weight vector of the function. Equations (3) and (4) represents the objective function of SVR, as follows:(3)$$L_{svr}=min\frac{1}{2}{\left|\omega \right|}^{2}+C\;\sum_{i=1}^{n}\left(\xi_{i}+\xi_{i}^{*}\right)$$
(4)$$s.t.\left\{\begin{array}{c} \left(\omega^{T}x_{i}+b\right)-y_{i}\leq \epsilon +\xi_{i}\\ y_{i}-\left(\omega^{T}x_{i}+b\right)\leq \epsilon +\xi_{i}^{*}\\ \;\;\;\;\;\;\;\xi_{i},\xi_{i}^{*}\;\;\;\;\;\;\;\geq 0 \end{array}\right.$$
where the positive constant, *C*, which is the regularization parameter, determines the flatness of the approximation function. $x_{i}\;\mathrm{and}\;y_{i}$ are the input and output variables of the *i*-th instance, respectively. ϵ is the error tolerance margin of the approximation function and $\xi_{i}\;\mathrm{and}\;\xi_{i}^{*}$ are slack variables for measuring the distance to the points outside the margin. The SVR input space computation can be performed using the kernel function, which returns the inner product of the input feature vectors, to solve the nonlinear problem by mapping lower-dimensional data into higher-dimensional data. This study used kernels in SVR, as follows (Equation (5)):(5)$$K\left(x_{i}\;,\;x_{j}\right)=\Phi \left(x_{i}\right)\cdot \;\Phi \left(x_{j}\right)$$
where $\Phi \left(x_{i}\right)\;\mathrm{and}\;\Phi \left(x_{j}\right)$ are feature space mapping functions.

Using the Lagrangian dual problem and kernel trick, SVR can be expressed as follows (Equation (6)):(6)$$y=\sum_{i=1}^{n}\left(\alpha_{i}^{*}-\alpha_{i}\right)K\left(x_{i}\;,\;x_{j}\right)+b$$
where $\alpha_{i}\;\mathrm{and}\;\alpha_{i}^{*}$ are Lagrange multipliers.

#### 2.4.2. Decision Tree

A decision tree is a decision-support-tree-like model formed of nodes and edges [19]. In the tree structure, class labels are represented by leaves and feature combinations are represented by branches. A decision tree splits nodes based on the result of the Gini impurity, which is a measure of diversity in a dataset (Equation (7)).
(7)$$G_{i}=1-\sum_{k=1}^{n}p_{i,\;\;k}^{2}$$
where $p_{i,\;k}^{2}$ is the proportion of samples belonging to class *k* for the *i*’th node.

#### 2.4.3. Random Forest Regression

Random forest regression is a bagging ensemble method of decision tree regression that is trained using the classification and regression tree (CART) algorithm. The objective function of CART is as follows (Equation (8)) [20]:(8)$$J(k,\;t_{k})=\frac{m_{left}}{m}G_{left}+\frac{m_{right}}{m}G_{right}$$
where $k\;\mathrm{and}\;t_{k}$ are the single feature and threshold, respectively; $G_{left/right}$ is the impurity of the subset; and $m_{left/right\;}$ is the number of samples of the subset. The ensemble method constructs multiple decision trees using a bagging algorithm known as bootstrap aggression. Each decision tree is trained by a sampling dataset with replacement and is aggregated by the average regression outcomes of the models. RF can mitigate the prediction variance and maintain unbiasedness as compared with a single decision tree.

#### 2.4.4. EXtreme Gradient Boost (XGBoost)

XGBoost is an ensemble algorithm that implements gradient-boosted decision trees [21]. Gradient boosting trains weak learners to create a strong ensemble model. Gradient boosting recursively adds a new decision tree model to correct the prior predictor model. Each decision tree was trained on the residual errors of the prior tree model. The sum of all of the prediction outcomes of the trees is the same as the ensemble prediction outcome.

#### 2.4.5. Deep Neural Network (DNN)

The DNN, which is composed of node layers, consists of an input layer, hidden layer, and output layer. Each node has a weight and threshold and is activated when the output of the node is above the specified threshold when using the activation function [22]. Batch normalization was used for each layer to avoid gradient vanishing or exploding. The model hyperparameters (the number of hidden layers and number of nodes in each layer) were determined by a grid search, and we determined the number of nodes and layers for the best estimation performance. The hidden layers were composed of three layers with 32, 64, and 32 nodes, respectively. A rectified linear unit was used for the activation function, the mean square error was the loss function used in the training, and Adam was used as an optimizer.

#### 2.4.6. Mixture Density Network (MDN)

The MDN, which is combined with a convolution network and mixture density model, models the mixture of parametric distributions, as shown in Equations (9) and (10) [23].
(9)$$p\left(y|x\right)=\sum_{i=1}^{n}\alpha_{i}\left(x\right)\Phi \left(y\;|\;\theta_{i}\right)$$
(10)$$s.t.\left\{\begin{array}{c} \overset{n}{\underset{i=1}{\sum }}\alpha_{i}\left(x\right)=1\\ \Phi \left(y\;|\;\theta_{i}\right)=\left(\mu_{i},\;\sigma_{i}^{2}\right) \end{array}\right.$$
where x and y are the input and output variables, respectively; *n* is the number of mixture components; and $\alpha_{i}\left(x\right)$ are mixing coefficients, which are prior probabilities (conditioned on *x*) corresponding to the mixture weight. $\Phi \left(y\;|\;\theta_{i}\right)$ is the conditional density composed of the mean ($\mu_{i}$) and variance ($\sigma_{i}^{2}$).

### 2.5. Model Evaluation

Using 30% of the total data, which were divided in the Bernoulli trial, the validation of the regression models was tested with the mean error and SEE, as shown in Equations (11) and (12).
(11)$$\mathrm{MAPE}\;\left(\%\right)=\frac{100}{N}\sum \left|\frac{{\hat{y}}_{i}-y_{i}}{y_{i}}\right|$$
(12)$$\mathrm{SEE}=\frac{\sum_{i=0}^{N}{\left({\hat{y}}_{i}-y_{i}\right)}^{2}}{N-2}$$
where $y_{i}\;\mathrm{and}\;{\hat{y}}_{i}$ are the measured and estimated values, respectively, and N  is the number of test samples.

## 3. Results

Detailed results of the regression model analysis are presented in Table 4 and Table 5. For each trained regression model, the coefficients of determination (R^2^), adjusted coefficients of determination, and SEE were used to analyze the estimated explanatory power of the regression models.

### 3.1. Performance Evaluation of the Regression Models

Table 4 presents a comparison of the FF variable estimation performance in the machine learning models. The DNN models presented the best performance with respect to R^2^ for estimation of the HGS (R^2^ = 0.622) and 30 s chair stand (R^2^ = 0.175), while the random forest model showed the best performance in the estimation of the chair sit-and-reach (R^2^ = 0.279), figure-of-eight walk (R^2^ = 0.381), and TUG (R^2^ = 0.212). For the estimation of the 2 min step test, the MDN model showed the most accurate estimation results (R^2^ = 0.119). Compared with the linear regression model [17], with the DNN model, R^2^ was improved by 3.7% and 1.2% in the HGS and 30 s chair stand estimation, respectively. It was also improved by 0.4% and 15.3% in estimation of the chair sit-and-reach and figure-of-eight walk, respectively, with the random forest model.

### 3.2. Performance Evaluation of the Regression Models without Outlier Data

Table 5 shows a comparison of the FF variable estimation performance in machine learning models without outlier data. In this performance evaluation, the outliers in the NFA datasets were removed using SRE to improve the training performance. Additionally, the Boruta algorithm and *p*-value were applied for feature selection of the input variables, as mentioned in Section 2.3. The input variables with a rank higher than 1 in the Boruta algorithm were excluded in the training (BMI and weight in TUG estimation). Furthermore, the input variables with a *p*-value higher than 0.05 (sex in the 2 min step test) were also excluded in the model training. The DNN-based regression model showed the best performance with respect to the R^2^ values for all FF variable estimations. Compared with the previous linear regression model [17], R^2^ was improved by 1.1%, 0.6%, 1.1%, 0.6%, 1%, and 1.4% for the HGS, 30 s chair, chair sit-and-reach, figure-of-eight walk, TUG, and 2 min step test with the DNN model. 

### 3.3. Regression Model Validity

Table 6 shows a comparison of the regression models’ validity with the test data, which is 30% of the total data. The mean absolute percentage error ranged from 0.084% to 22.68% in the regression models (DNN model, HGS: MAPE = 0.16% and SEE = 4.135 kg, 30 s chair stand test: MAPE = 0.205% and SEE = 4.169 times, chair sit-and-reach test: MAPE = 20.92% and SEE = 6.228 cm, figure-of-eight walk test: MAPE = 0.097% and SEE = 3.546 s, TUG test: MAPE = 0.084% and SEE = 0.805 s, and 2 min step test: MAPE = 0.099% and SEE = 13.00 times). Figure 2 shows the relationship between the measured and predicted FF variables using scatter plots.

## 4. Discussion

FF variables, which can be used as an index of healthcare, have been used to assess the health conditions of older adults, and several researchers have studied the correlation between independent variables and FF variables. In previous studies, MLR was used to develop a prediction model for the FF variables. However, MLR, which cannot represent the nonlinearity of data, has limitations in estimating FF variables. In addition, prior studies focused on predicting a specific group’s superior physical functionality, such as that of athletes, which is not appropriate for the prediction of FF variables in older adults. The present study focused on developing a regression model for estimating the FF variables of older adults in Korea with easy-to-measure independent variables. To obtain an accurate regression model, we compared various machine learning and deep learning regression models. This study demonstrated the highest performance of the DNN model in FF variable estimation compared with the other regression models. With the developed regression model, it would be helpful to monitor the FF in older adults in daily life.

The correlation coefficient shown in Table 2 represents the strength and direction of the linear relation between the input and output variables. In a previous study, height, weight, and BMI were significantly correlated with HGS for older adults [24]. In this study, HGS had a higher correlation coefficient with these independent variables, and presented the most accurate estimation results compared with the other FF variables. From these results, we can infer that it is important to select input variables with a strong correlation in order to obtain higher estimation results. 

Using nonlinear regression models, we focused on predicting the FF variables of older adults using independent variables. The mean explanatory power of HGS was high in the HGS and DNN regression models (MLR: 61.4%, SVM: 62.1%, RF: 61.9%, XGBoost: 62.0%, DNN: 62.2%, and MDN: 61.7%). In this study, outlier removal and feature selection were conducted. The mean explanatory power of HGS without outlier data was 78.4%, which was the highest value in the DNN model (MLR: 77.3%, SVM: 78.4%, RF: 74.2%, XGBoost: 78.3%, DNN: 78.4%, MDN: 78.3%). Our proposed regression model’s explanatory power of HGS was improved by approximately 25% compared with previous studies [25,26]. In our previous study, we developed a linear regression model for predicting FF variables of South Korean older adults [17]. However, the previous study did not cover the nonlinearity of the dataset and only used multiple linear regression models without considering other regression models, which may likely improve the prediction accuracy. Hence, we tested various regression models covering data nonlinearity and proposed the best performance regression model. The DNN-based regression model had a better performance than the linear regression model. Comparing the model’s validation, SEE was improved by 16.6% in HGS, 28.2% in 30 s chair stand, 25.9% in chair sit-and-reach, 50.1% in figure-of-eight walk, 56.7% in TUG, and 48.5% in the 2 min step test.

The coefficient of determination of the proposed model was too low, making it insufficient for practical applications, except for predicting HGS. The coefficients of determination in the 30 s chair stand (adjusted R^2^ = 0.300), chair sit-and-reach (adjusted R^2^ = 0.441), figure-of-eight walk (adjusted R^2^ = 0.395), TUG (adjusted R^2^ = 0.389), and 2 min step tests (adjusted R^2^ = 0.207) were in the mid-range. It was inferred that more input variables were required to analyze the relationship with the FF variables. Hence, additional variables, such as the individual physical activity level or nutrition, which are correlated with the FF variables [27], were needed to improve the prediction accuracy. Moreover, we used the general older adults’ independent variables, which did not contain their health status, such as personal physical illness/disease information, even though these might be correlated with the FF variables. Chronic diseases, such as cardiovascular disease and type 2 diabetes, cause mortality in older adults [28]. Information obtained from blood pressure measurements and blood glucose tests could be used as input variables to predict the correlation with FF variables. These parameters may also be used as indicators to isolate the effects of physical illness/disease information. The DNN-based regression model showed the highest performance for most of the FF variables, but the amount of improvement was <1.6% compared with the other regression models in validation. Selecting machine learning models with a computational efficiency is considered practical for predicting HGS.

## 5. Conclusions

Herein, we proposed an FF variable prediction model based on machine learning and deep learning regression with easy-to-measure independent variables, and compared the performance of each model. This study demonstrated a correlation between older adults’ independent variables and the FF variables, especially HGS. However, the estimation results of the FF variables, except for HGS, were unsatisfactory for monitoring older adults’ physical functionality and providing personalized workout programs. The results showed the difficulty in predicting the FF variables using six independent variables (age, sex, height, weight, percent body fat, and BMI), which were insufficient for representing the correlation of FF variables. In future research, additional variables, including the physical activity level and nutritional status, will be used to enhance the accuracy of the estimation results.

## Figures and Tables

**Figure 1 ijerph-19-09754-f001:**
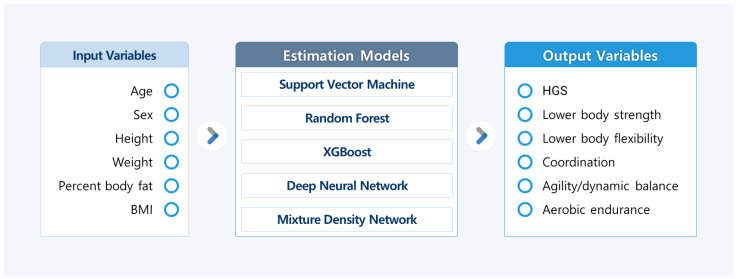
Summary of various regression models to predict FF variables with independent variables. Estimation models predicted each output variable. FF, functional fitness; BMI, body mass index; HGS, hand grip strength; XGBoost, extreme gradient boosting.

**Figure 2 ijerph-19-09754-f002:**
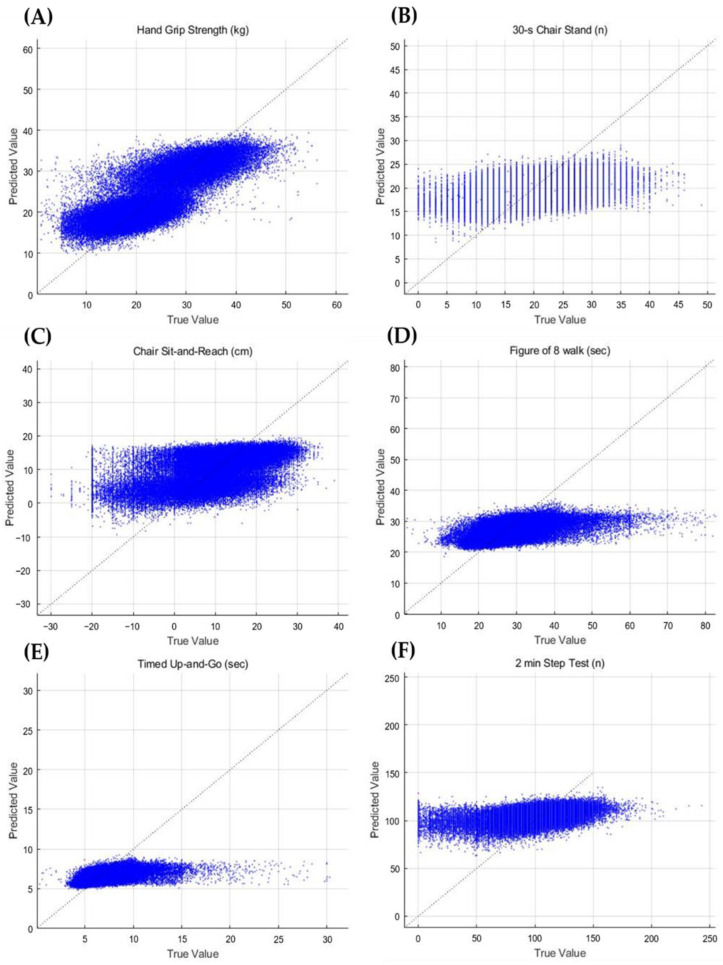
Relationship between the measured and predicted FF variables using scatter plots. (**A**) HGS, (**B**) 30 s chair stand test, (**C**) chair sit-and-reach test, (**D**) figure-of-eight walk test, (**E**) timed up-and-go test, and (**F**) 2 min step test results.

**Table 1 ijerph-19-09754-t001:** Summary of the NFA dataset used to train the FF variable estimation model.

Data Type	Variables	Training Dataset		Validation Dataset	
		Men (*n* = 42,991)	Women (*n* = 82,281)	Men (*n* = 18,474)	Women (*n* = 35,214)
Independent Variable (Input)	Age (year)	73.26 ± 5.45	72.55 ± 5.62	73.31 ± 5.45	72.51 ± 5.59
	Height (cm)	165.10 ± 5.86	152.37 ± 5.54	165.10 ± 5.79	152.38 ± 5.52
	Weight (kg)	66.33 ± 8.87	57.54 ± 8.00	66.39 ± 8.84	57.49 ± 7.93
	Percent body fat (%)	24.31 ± 2.80	24.77 ± 3.13	24.33 ± 2.79	24.75 ± 3.09
	BMI (kg/m^2^)	26.01 ± 6.39	34.97 ± 6.42	26.06 ± 6.42	34.93 ± 6.38
Functional Fitness Variable (Output)	HGS (kg)	30.76 ± 6.66	19.45 ± 4.84	30.75 ± 6.65	19.49 ± 4.82
	30-s chair stand (n)	20.58 ± 6.40	18.23 ± 6.05	20.53 ± 6.40	18.27 ± 6.05
	Chair sit-and-reach (cm)	3.87 ± 9.70	13.06 ± 8.06	3.79 ± 9.64	13.12 ± 8.02
	Figure of 8 walk (s)	26.04 ± 7.01	28.00 ± 7.96	26.13 ± 7.16	27.95 ± 7.90
	Timed up-and-go (s)	6.20 ± 1.81	6.79 ± 2.07	6.20 ± 1.78	6.77 ± 2.02
	2-sim step test (n)	107.20 ± 24.90	100.31 ± 27.49	107.02 ± 24.50	100.58 ± 27.18

NFA, National Fitness Award; BMI, body mass index; HGS, hand grip strength.

**Table 2 ijerph-19-09754-t002:** Pearson’s correlation analysis between the independent variables and FF variables.

	HGS	30-s Chair Stand	Chair Sit-and-Reach	Figure of 8 Walk	Timed Up-and-Go	2-min Step Test
Age	−0.223	−0.317	−0.250	0.429	0.397	−0.306
Sex	−0.693	−0.172	0.454	0.122	0.144	−0.126
Height	0.688	0.169	−0.307	−0.223	−0.239	0.196
Weight	0.492	0.029	−0.229	−0.070	−0.082	0.074
Percent body fat	−0.466	−0.246	0.153	0.212	0.212	−0.189
BMI	0.013	−0.111	−0.016	0.106	0.104	−0.079

FF, functional fitness; Sex, male is expressed as 1 and female is expressed as 2.

**Table 3 ijerph-19-09754-t003:** *p*-values and Boruta feature selection of each independent variable.

	HGS		30 s Chair Stand		Chair Sit-and-Reach		Figure of 8 Walk		Timed Up-and-Go		2 min Step Test	
	*p*-Value	Rank	*p*-Value	Rank	*p*-Value	Rank	*p*-Value	Rank	*p*-Value	Rank	*p*-Value	Rank
Age	0.000	1	0.000	1	0.000	1	0.000	1	0.000	1	0.000	1
Sex	0.000	1	0.000	1	0.000	1	0.000	1	0.000	1	0.554	1
Height	0.000	1	0.000	1	0.000	1	0.000	1	0.000	1	0.000	1
Weight	0.000	1	0.000	1	0.000	1	0.000	1	0.000	3	0.000	1
Percent body fat	0.000	1	0.000	1	0.000	1	0.000	1	0.000	1	0.000	1
BMI	0.000	1	0.000	1	0.000	1	0.034	1	0.059	2	0.000	1

**Table 4 ijerph-19-09754-t004:** Comparison of the estimated regression models predicting the FF variables.

Support Vector Regression	R	R^2^	Adjusted R^2^	SEE
HGS	0.788	0.621	0.621	4.750 kg
30 s chair stand	0.417	0.174	0.174	5.635 n
Chair sit-and-reach	0.515	0.265	0.265	8.395 cm
Figure-of-eight walk	0.423	0.179	0.179	6.784 s
Timed up-and-go	0.391	0.153	0.153	1.846 s
2 min step test	0.313	0.098	0.098	25.73 n
Random Forest	R	R^2^	Adjusted R^2^	SEE
HGS	0.787	0.619	0.619	4.681 kg
30 s chair stand	0.406	0.165	0.165	5.293 n
Chair sit-and-reach	0.528	0.279	0.279	8.124 cm
Figure-of-eight walk	0.617	0.381	0.381	3.116 s
Timed up-and-go	0.460	0.212	0.212	1.198 s
2 min step test	0.310	0.096	0.096	22.63 n
XGBoost	R	R^2^	Adjusted R^2^	SEE
HGS	0.787	0.620	0.620	4.755 kg
30 s chair stand	0.409	0.167	0.167	5.660 n
Chair sit-and-reach	0.521	0.272	0.272	8.355 cm
Figure-of-eight walk	0.442	0.195	0.195	6.716 s
Timed up-and-go	0.416	0.173	0.173	1.823 s
2 min step test	0.321	0.103	0.103	25.66 n
DNN	R	R^2^	Adjusted R^2^	SEE
HGS	0.789	0.622	0.622	4.741 kg
30 s chair stand	0.418	0.175	0.175	5.640 n
Chair sit-and-reach	0.523	0.274	0.274	8.347 cm
Figure-of-eight walk	0.449	0.202	0.202	6.688 s
Timed up-and-go	0.423	0.179	0.179	1.817 s
2 min step test	0.329	0.108	0.108	25.59 n
MDN	R	R^2^	Adjusted R^2^	SEE
HGS	0.785	0.617	0.617	4.771 kg
30 s chair stand	0.394	0.155	0.155	5.700 n
Chair sit-and-reach	0.522	0.273	0.273	8.349 cm
Figure-of-eight walk	0.448	0.201	0.201	6.693 s
Timed up-and-go	0.436	0.190	0.190	1.804 s
2 min step test	0.345	0.119	0.119	25.44 n

FF, functional fitness; SEE, standard error of estimation; XGBoost, extreme gradient boosting; DNN, deep neural network; MDN, mixture density network.

**Table 5 ijerph-19-09754-t005:** Comparison of the estimated regression models predicting the FF variables without outlier data.

FF Variables	SRE		Independent Variables (Input Variables)	
HGS	SRE 32: *n* = 101,438		Age, Sex, Height, Weight, Percent body fat, BMI	
30 s chair stand	SRE 39: *n* = 102,726		Age, Sex, Height, Weight, Percent body fat, BMI	
Chair sit-and-reach	SRE 35: *n* = 102,640		Age, Sex, Height, Weight, Percent body fat, BMI	
Figure-of-eight walk	SRE 22: *n* = 79,724		Age, Sex, Height, Weight, Percent body fat, BMI	
Timed up-and-go	SRE 36: *n* = 94,621		Age, Sex, Height, Percent body fat	
2-min step test	SRE 28: *n* = 91,420		Age, Height, Weight, Percent body fat, BMI	
Support Vector Regression	R	R^2^	Adjusted R^2^	SEE
HGS	0.885	0.784	0.784	3.069 kg
30-s chair stand	0.548	0.300	0.300	3.800 n
Chair sit-and-reach	0.664	0.441	0.441	5.436 cm
Figure-of-eight walk	0.628	0.395	0.395	3.083 s
Timed up-and-go	0.624	0.389	0.389	0.705 s
2-min step test	0.455	0.207	0.207	12.46 n
Random Forest	R	R^2^	Adjusted R^2^	SEE
HGS	0.861	0.742	0.742	3.336 kg
30-s chair stand	0.514	0.264	0.264	5.492 n
Chair sit-and-reach	0.655	0.429	0.429	3.197 cm
Figure-of-eight walk	0.590	0.348	0.348	3.197 s
Timed up-and-go	0.588	0.346	0.346	0.729 s
2-min step test	0.438	0.192	0.192	12.57 n
XGBoost	R	R^2^	Adjusted R^2^	SEE
HGS	0.885	0.783	0.783	3.069 kg
30-s chair stand	0.548	0.301	0.301	3.800 n
Chair sit-and-reach	0.662	0.438	0.438	5.448 cm
Figure-of-eight walk	0.628	0.395	0.395	3.080 s
Timed up-and-go	0.626	0.392	0.392	0.704 s
2-min step test	0.453	0.205	0.205	12.48 n
DNN	R	R^2^	Adjusted R^2^	SEE
HGS	0.885	0.784	0.784	3.054 kg
30-s chair stand	0.550	0.302	0.302	3.794 n
Chair sit-and-reach	0.668	0.446	0.446	5.418 cm
Figure-of-eight walk	0.629	0.396	0.396	3.078 s
Timed up-and-go	0.628	0.394	0.394	0.702 s
2-min step test	0.458	0.210	0.210	12.44 n
MDN	R	R^2^	Adjusted R^2^	SEE
HGS	0.885	0.783	0.783	3.069 kg
30-s chair stand	0.529	0.280	0.280	3.852 n
Chair sit-and-reach	0.646	0.417	0.417	5.552 cm
Figure-of-eight walk	0.628	0.394	0.394	3.083 s
Timed up-and-go	0.622	0.387	0.387	0.707 s
2-min step test	0.451	0.203	0.203	12.49 n

SRE, studentized residual; SEE, standard error of estimation.

**Table 6 ijerph-19-09754-t006:** Validation of estimating accuracy.

	HGS (kg)		30 s Chair Stand (n)		Chair Sit-and-Reach (cm)		Figure of 8 Walk (s)		Timed Up-and-Go (s)		2 min Step Test (n)	
	MAPE (%)	SEE	MAPE (%)	SEE	MAPE (%)	SEE	MAPE (%)	SEE	MAPE (%)	SEE	MAPE (%)	SEE
MLR	0.160	4.216	0.206	4.214	20.12	6.315	0.100	3.565	0.089	0.822	0.100	13.13
SVR	0.157	4.147	0.202	4.183	19.81	6.274	0.098	3.549	0.087	0.817	0.099	13.03
RF	0.160	4.258	0.210	4.307	20.75	6.455	0.102	3.649	0.091	0.840	0.103	13.56
XGBoost	0.158	4.160	0.205	4.178	19.96	6.252	0.098	3.529	0.087	0.814	0.099	13.06
DNN	0.157	4.135	0.205	4.169	20.92	6.228	0.097	3.546	0.084	0.805	0.099	13.00
MDN	0.158	4.141	0.214	4.228	22.68	6.313	0.096	3.517	0.086	0.831	0.099	13.03

MAPE, mean absolute percentage error; MLR, multiple linear regression; SVR, support vector regression; RF, random forest; SEE, standard error of estimation.

## Data Availability

Restrictions apply to the availability of data. Data were obtained from the Korea Sports Promotion Foundation.

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
