# Peer review of "Estimation of Functional Fitness of Korean Older Adults Using Machine Learning Techniques: The National Fitness Award 2015–2019"

_ijerph, 2022, doi:10.3390/ijerph19159754_

Round 1

Reviewer 1 Report

The article “Estimation of Functional Fitness of Korean Older Adults Using Machine Learning Techniques: The National Fitness Award 2015–2019” by Sang-Hun Lee and co-authors proposed an FF variable prediction model based on machine learning and deep learning regression with easy-to-measure independent variables and compared the performance of each model. Although the results of the FF variables, except for HGS, were unsatisfactory for monitoring older adults’ physical functionality, the presented data could be accepted for publication in IJERPH. Please follow the comments below.

(1) In line 34, please include references to support the statement.

(2) In Line 104 and 105, follow one format to represent the percentage value.

(3) Please describe the background of The National Fitness Award 3 2015–2019 in the main manuscript.

(4) Physical illness/disease information, pretreatment and post-treatment information might significantly contribute to this correlative estimation. What could be an excellent way to isolate the effect of each circumstance? Please include in the main manuscript?

Author Response

These responses are for the manuscript ijerph-1836747 entitled "Estimation of Functional Fitness of Korean Older Adults Using Machine Learning Techniques: The National Fitness Award 2015-2019" submitted to International Journal of Environmental Research and Public Health.

The authors are grateful for the reviewers' constructive comments regarding the first manuscript. We have done our best to answer all the questions raised by the reviewers, and have revised the manuscript per their advices. We have quoted below all the reviewers' comments in order, and put our answers (red) and revised script in the paper (blue) after each comment.

Comments of the Reviewer 1

[Comment 1.1]

The article "Estimation of Functional Fitness of Korean Older Adults Using Machine Learning Techniques: The National Fitness Award 2015–2019" by Sang-Hun Lee and co-authors proposed an FF variable prediction model based on machine learning and deep learning regression with easy-to-measure independent variables and compared the performance of each model. Although the results of the FF variables, except for HGS, were unsatisfactory for monitoring older adults' physical functionality, the presented data could be accepted for publication in IJERPH. Please follow the comments below.

[Response 1.1]

Thank you for your comments. We have done our best to respond to your comments and revised the manuscript as per your suggestions.

[Comment 1.2]

In line 34, please include references to support the statement.

[Response 1.2]

Thank you for your constructive comment. We have included ‘The world population prospects 2022: Summary of Results from UN’ as a reference to support the statement in line 34.

[1] United Nations Department of Economic and Social Affairs, Population Division (2022). World Population Prospects 2022: Summary of Results. UN DESA/POP/2022/TR/NO. 3

[Comment 1.3]

In Line 104 and 105, follow one format to represent the percentage value.

[Response 1.3]

Thank you for your constructive comment. We corrected this mistake by changing the 'seven percentage' to '70%' in the revised manuscript as follows:

70% of the data (total: n=125,272, men: n=42,911, women: n=82,281) was used as the training dataset, and 30% of the data (total: n=53,688; men: n=18,474; women: n=35,214) was used as the validation dataset.

[Comment 1.4]

Please describe the background of The National Fitness Award 3 2015–2019 in the main manuscript.

[Response 1.4]

Thank you for your constructive comment. We have included a specific explanation about the background of the National Fitness Award in the main manuscript as follows:

The National Fitness Award (NFA) is a program carried out by the Ministry of Culture, Sports, and Tourism (MCST) and Korea Sports Promotion Foundation (KSPF) to measure physical fitness levels of general Koreans aiming to help people live healthier. This paper employes the NFA datasets of elderly adults (age: ≥65 years), gathered by the KSPF to train the machine learning based estimation models.

[Comment 1.5]

Physical illness/disease information, pretreatment and post-treatment information might significantly contribute to this correlative estimation. What could be an excellent way to isolate the effect of each circumstance? Please include in the main manuscript?

[Response 1.5]

Thank you for your constructive comment. The personal conditions including physical illness/disease and pre/post-treatment may contribute to the individuals’ FF variables as the reviewer commented. Unfortunately, the estimation model proposed in this paper cannot reflect the effect of those personal conditions because it does not include those informations in the input variables. Actually, the aim of this study is to develop the estimation model of FF variable for general elderly adults based on the NFA datasets which does not includes desease informations. At this time, we cannot isolate the effect of individual’s each circumstance in the FF estimation. Physical illness and diseases may negatively affect to the accuracy of the proposed estimation model. It is one of limitations of the proposed model. We have discussed this limiation and proposed to include additional input variables in the estimation model to reflect various personal circumstances such as illness and disease conditions as a future work as follows:

Moreover, we used general older adults’ independent variables, which does not contain health status such as personal physical illness/disease information even though these might be correlated with the FF variables. Chronic diseases, such as cardiovascular dis-ease and type 2 diabetes, cause mortality in older adults [29]. Information obtained from blood pressure measurements and blood glucose tests can be used as input variables to predict the correlation with FF variables. These parameters may also be used as indicators to isolate the effects of physical illness/disease information.

Reviewer 2 Report

The paper of Lee et al., entitled “Estimation of Functional Fitness of Korean Older Adults Using Machine Learning Techniques: The National Fitness Award 3 2015–2019” analyses large dataset obtained for older Korean adults aged >65 years. It is based on measurements of functional fitness (FF) to track the decline in physical abilities. Six FF variables (hand grip strength (HGS), lower body strength (30-second chair stand), 19 lower body flexibility (chair sit-and-reach test), coordination (figure-of-8 walk test), agility/dynamic 20 balance (timed up-and-go test), and aerobic endurance (2-minute step test) were tested and different machine learning and deep learning regression models were compared. The paper is structured in a clear and logical way, the results presentation is flawless and the discussion of the data is straightforward. The DNN model in FF variable estimation compared to other regression models was found to have the highest performance. The authors also comment on the limitations of the applied approach and suggest the addition of other variables in order to perfect the analysis.

I recommend this manuscript to be published as it is in Int. J. Environ. Res. Public Health.

Please find below my specific comments regarding the manuscript.

1.
To precisely measure functional fitness (FF) in the elderly population without the utilization of expensive equipment and software products is a major concern in the modern world. In this respect the authors question the utilization of machine learning/deep learning algorithms in order to develop a prediction model for FF variables (hand grip strength, lower body strength, lower body flexibility, coordination, agility/dynamic balance, and aerobic endurance) based on easy-to-measure physical variables (age, sex, height, weight, percent body fat, BMI).

2.
The topic is original and timely since the number of elderly people over 65 years is a major problem throughout the developed world, associated with a number of social and medical problems. The FF is tightly related to the quality of life and therefore easy to apply methods for its determination would benefit the elderly people and the medical and social workers.

3.
Previous studies have utilized multiple linear regression analyses for FF determination, which only reflects linear relation between the inputs and outputs. Non-linear relations were also considered; however, they were utilizing data obtained for well-trained individuals which is not representative for a general elderly population. The current manuscript utilizes The National Fitness Award (NFA) datasets obtained for adults aged ≥65 years, which are obtained in 75 different sites throughout the Republic of Korea and no requirement for physical training is applied.

4.
The applied methodology fits the aim of the manuscript. However, there are some minor points to be addressed regarding the Results presentation.

Heading of Table 2 – Specify that Pearson’s correlation analysis is performed.

3.1 Performance Evaluation of the Regression Models”- it seems that in this paragraph only DNN model is discussed. Also the values in text and in Table 4 do not coincide, please double-check.

3.2 Performance Evaluation of the Regression Models Without Outlier Data” – why only DNN is discussed?

Table 5: Why “Weight” and “BMI” are not included as independent variables for “Timed up-and-go”?

As the authors themselves conclude, the addition of other variables (such as physical activity level and nutritional status) would improve the proposed model.

5.
The conclusion is consistent with the presented data. The authors state that high correlation is established for Hang-grip test and the measured physical parameters, however the correlation for the other FF and physical variables is not satisfactory. They discuss the need to add more physical parameters in the analysis in order to perfect it to an extent that can have practical realization.

6. Comprehensive citation of relevant works is presented.

7. Addition of explanation of the SEE abbreviation in the description of Tables 5 and 6 would be helpful.

Author Response

These responses are for the manuscript ijerph-1836747 entitled "Estimation of Functional Fitness of Korean Older Adults Using Machine Learning Techniques: The National Fitness Award 2015-2019" submitted to International Journal of Environmental Research and Public Health.

The authors are grateful for the reviewers' constructive comments regarding the first manuscript. We have done our best to answer all the questions raised by the reviewers, and have revised the manuscript per their advices.

Comments of the Reviewer 2

[Comment 2.1]

The paper of Lee et al., entitled "Estimation of Functional Fitness of Korean Older Adults Using Machine Learning Techniques: The National Fitness Award 3 2015–2019" analyses large dataset obtained for older Korean adults aged >65 years. It is based on measurements of functional fitness (FF) to track the decline in physical abilities. Six FF variables (hand grip strength (HGS), lower body strength (30-second chair stand), 19 lower body flexibility (chair sit-and-reach test), coordination (figure-of-8 walk test), agility/dynamic 20 balance (timed up-and-go test), and aerobic endurance (2-minute step test) were tested and different machine learning and deep learning regression models were compared. The paper is structured in a clear and logical way, the results presentation is flawless and the discussion of the data is straightforward. The DNN model in FF variable estimation compared to other regression models was found to have the highest performance. The authors also comment on the limitations of the applied approach and suggest the addition of other variables in order to perfect the analysis.

I recommend this manuscript to be published as it is in Int. J. Environ. Res. Public Health.

Please find below my specific comments regarding the manuscript.

[Response 2.1]

We appreciate your comments. We have done our best to respond to your comments and revised the manuscript as per your suggestions.

[Comment 2.2]

To precisely measure functional fitness (FF) in the elderly population without the utilization of expensive equipment and software products is a major concern in the modern world. In this respect the authors question the utilization of machine learning/deep learning algorithms in order to develop a prediction model for FF variables (hand grip strength, lower body strength, lower body flexibility, coordination, agility/dynamic balance, and aerobic endurance) based on easy-to-measure physical variables (age, sex, height, weight, percent body fat, BMI).

[Response 2.2]

We appreciate your comments.

[Comment 2.3]

The topic is original and timely since the number of elderly people over 65 years is a major problem throughout the developed world, associated with a number of social and medical problems. The FF is tightly related to the quality of life and therefore easy to apply methods for its determination would benefit the elderly people and the medical and social workers.

[Response 2.3]

We appreciate your comments.

[Comment 2.4]

Previous studies have utilized multiple linear regression analyses for FF determination, which only reflects linear relation between the inputs and outputs. Non-linear relations were also considered; however, they were utilizing data obtained for well-trained individuals which is not representative for a general elderly population. The current manuscript utilizes The National Fitness Award (NFA) datasets obtained for adults aged ≥65 years, which are obtained in 75 different sites throughout the Republic of Korea and no requirement for physical training is applied.

[Response 2.4]

We appreciate your comments.

[Comment 2.5]

The applied methodology fits the aim of the manuscript. However, there are some minor points to be addressed regarding the Results presentation.

Heading of Table 2 – Specify that Pearson's correlation analysis is performed.

[Response 2.5]

Thank you for your constructive comment. We have revised the heading of Table 2 according to your comments as follows:

Table 2. Pearson's correlation analysis between the independent variables and FF variables.

[Comment 2.6]

"3.1 Performance Evaluation of the Regression Models"- it seems that in this paragraph only DNN model is discussed. Also the values in text and in Table 4 do not coincide, please double-check.

"3.2 Performance Evaluation of the Regression Models Without Outlier Data" – why only DNN is discussed?

[Response 2.6]

Thank you for your constructive comment. In the first draft, we presented the results of the estimation model focusing on DNN model since it shows the best performance in most of FF estimation. According to reviewer’s comments, we revised the description of the results to presents the performance of not only the DNN model but also the other machine learning models. And also we added desriptions that compares the performance of the machine learning models with that of the linear regression model. Furthermore, we carefully checked the mismatched values in the text and revised them.

3.1 Performance Evaluation of the Regression Models

Table 4 presents a comparison of FF variable estimation performance in machine learning models. The DNN models presented the best performance with respect to the R2  for estimation of HGS (R2 = 0.622) and 30-s chair stand (R2 = 0.175) while the random forest model shows the best performance in estimation of chair sit-and-reach (R2 = 0.279), figure of 8 walk (R2 = 0.381) and TUG (R2 = 0.212). For the estimation of 2-min step test, the MDN model shows the most accurate estimation results (R2 = 0.119). Compared with the linear regression model [17], R2 was improved by 3.7% and 1.2% in HGS and 30-s chair stand estimation respectively with the DNN model. It is also improved by 0.4% and 15.3% in estimation of chair sit-and-reach and figure of 8 walk respectively with the random forest model.

3.2 Performance Evaluation of the Regression Models Without Outlier Data

Table 5 shows a comparison of FF variable estimation performance in machine learning models without outlier data. In this performance evaluation, the outliears in the NFA datasets were removed using SRE to improve training performance. Additionally, the Boruta algorithm and p-value were applied for the feature selection of input variables, as mentioned in section 2.3. The input varialbes whose rank is higher than 1 in Boruta algorithm were excluded in the training (BMI and weight in TUG estimation). Furthermore, the input variables whose p-value is higher than 0.05 (sex in the 2-min step test) were also excluded in the model training. The DNN-based regression model shows the best performance with respect to R2 values for all FF variable estimation. Compared with the previous linear regression model [17], R2 was improved by 1.1%, 0.6%, 1.1%, 0.6%, 1%, and 1.4% for HGS, 30-s chair, chair sit-and-reach, figure of 8 walk, TUG, and 2-min step test with the DNN model.

[Comment 2.7]

Table 5: Why "Weight" and "BMI" are not included as independent variables for "Timed up-and-go"?  As the authors themselves conclude, the addition of other variables (such as physical activity level and nutritional status) would improve the proposed model.

[Response 2.7]

Thank you for your constructive comment. The input vaiables which is not correlated to the output may degrade the estimation performance. In this paper, we used feature selection methods using p-value and the Boruta algorithm to evaluate the degree of correlation between each input and output variables. The results of Boruta algorithm shown in Table 3 presented that BMI and weight ranked 2 and 3 respectivley. It means that the correlation of these variables with outputs are lower than other variables. We excluded these two variables in the TUG estimation to enhance the performance of the estimation.

In this performance evaluation, the outliears in the NFA datasets were removed using SRE to improve training performance. Additionally, the Boruta algorithm and p-value were applied for the feature selection of input variables, as mentioned in section 2.3. The input varialbes whose rank is higher than 1 in Boruta algorithm were excluded in the training (BMI and weight in TUG estimation). Furthermore, the input variables whose p-value is higher than 0.05 (sex in the 2-min step test) were also excluded in the model training.

[Comment 2.8]

The conclusion is consistent with the presented data. The authors state that high correlation is established for Hang-grip test and the measured physical parameters, however the correlation for the other FF and physical variables is not satisfactory. They discuss the need to add more physical parameters in the analysis in order to perfect it to an extent that can have practical realization.

[Response 2.8]

We appreciate your comments.

[Comment 2.9]

Comprehensive citation of relevant works is presented.

[Response 2.9]

We appreciate your comments.

[Comment 2.10]

Addition of explanation of the SEE abbreviation in the description of Tables 5 and 6 would be helpful.

[Response 2.10]

Thank you for your constructive comment. We have included an additional explanation of the SEE in Tables 5 and 6.

(In Table 5 and 6) SEE, standard error of estimation.

Reviewer 3 Report

General

First of all, the reviewer would like to thank the authors for their work and efforts in trying to improve  sports science  healthknowledge.

The authors proposed an FF variable prediction model using easy to-measure physical independent variables in Korean older adults aged >65 years. The article is well written. However, I suggest some points for improvement

Abstract:

They should clearly state the aim of the study

Introduction:

Line 44: the authors indicate that the WHO establishes a minimum of 150 min of high-intensity exercise. This is correct but incomplete. The WHO recommends a minimum of 150-300 min of moderate exercise or 150 min of intense exercise, or a combination of both. This is for aerobic exercise. The WHO also recommends a minimum of 2 days per week of strength exercise. Authors should complete this information according to current WHO guidelines.

The last paragraph of the introduction should clearly state the aim of the study. Statistical aspects that are not necessary to establish the aim should be reported in the material and methods section.

Material and methods:

- Table 2. Indicate the significance level for the correlation analysis. Authors should indicate in the table which variables obtain a linear correlation.

Results

Well written

Discussion:

I recommend extending the discussion with previous evidence of the relationship between the variables analysed.

References:

No coments

Author Response

Responses to the Reviewers' Comments

These responses are for the manuscript ijerph-1836747 entitled "Estimation of Functional Fitness of Korean Older Adults Using Machine Learning Techniques: The National Fitness Award 2015-2019" submitted to International Journal of Environmental Research and Public Health.

The authors are grateful for the reviewers' constructive comments regarding the first manuscript. We have done our best to answer all the questions raised by the reviewers, and have revised the manuscript per their advices.

Comments of the Reviewer 3

[Comment 3.1]

First of all, the reviewer would like to thank the authors for their work and efforts in trying to improve  sports science  health knowledge. The authors proposed an FF variable prediction model using easy to-measure physical independent variables in Korean older adults aged >65 years. The article is well written. However, I suggest some points for improvement

[Response 3.1]

Thank you for your comments. We have done our best to respond to your comments and revised the manuscript as per your suggestions.

[Comment 3.2]

Abstract:

They should clearly state the aim of the study

[Response 3.2]

Thank you for your constructive comment. The main contributions of our study include the proposition of an FF variable prediction model to monitor older adults' FF variables easily and deriving an optimal prediction model by testing various machine learning / deep learning models. We revised the abstract to cleary state the aim of the study as follows:

Abstract: Measuring functional fitness (FF) to track the decline in physical abilities is important to maintain a healthy life in old age. This paper aims to develop an estimation model of FF vari-ables, which represents strength, flexibility, and aerobic endurance, using easy-to-measure phys-ical parameters for Korean older adults aged over 65 years. The estimation models were devel-oped using various machine learning techniques and trained with the National Fitness Award datasets from 2015 to 2019 compiled by the Korea Sports Promotion Foundation. The machine learning based nonlinear regression models were employed to improve the performance of the previous linear regression models. To derive the optimal estimation model which shows the best estimation accuracy, we developed five different machine learning-based estimation mod-els and compares the estimation accuracy not only among the machine learning models but also with the previous linear regression model. The coefficient of determination of the FF variables was used to compare the performance of each model; the mean absolute percentage error (MAPE) and standard error of estimation (SEE) were used to evaluate the models’ performance. The deep neural network (DNN) model presented the best performance among the regression models in estimation of all FF variables. The coefficient of determination in the HGS test was 0.784 while those of the others were less than 0.5 meanin that HGS of older adults can be reliably estimated using easy-to-measure independent variables.

[Comment 3.3]

Introduction:

Line 44: the authors indicate that the WHO establishes a minimum of 150 min of high-intensity exercise. This is correct but incomplete. The WHO recommends a minimum of 150-300 min of moderate exercise or 150 min of intense exercise, or a combination of both. This is for aerobic exercise. The WHO also recommends a minimum of 2 days per week of strength exercise. Authors should complete this information according to current WHO guidelines.

[Response 3.3]

Thank you for your constructive comment. We have revised the information according to the current 2020 WHO guidelines on physical activity and sedentary behavior. As per the WHO guidelines, we have included information on aerobic exercise, strength exercise, and multi-component physical activities as follows.

The World Health Organization (WHO) recommends that elderly individuals perform moderate-intensity aerobic exercises for at least 150–300 min per week, vigorous-intensity aerobic exercise for at least 75–150 min, or a combination of both. The WHO also recom-mends strength training for more than 2 days a week and multicomponent physical activ-ities more than 3 days a week [7].

[Comment 3.4]

The last paragraph of the introduction should clearly state the aim of the study. Statistical aspects that are not necessary to establish the aim should be reported in the material and methods section.

[Response 3.4]

Thank you for your constructive comment. We have included the aims and the main contributions of our study and excluded the statistical aspects in the last paragraph of the introduction as follows.

In this paper, we proposes machine learning-based estimation model for FF variables with easy-to-measure physical variables in Korean older adults. To derive the optimal estimation model which shows the best estimation performance, we have constructed various machine learning-based estimation models and evaluated the estimation performance of each model.

The main contributions of this study are as follows:

  1. This study proposed the FF variable estimation model for evaluating the physical fitness level of elderly adults using easy-to-measure independent variables. The proposed model can be used as effective tools to evaluate personal fitness level in smart fitness services.
  2. Various nonlinear machine learning regression models were constructed and evaluated to compare the accuracy with the previous linear model and derive the optimal estimation model presenting the best estimation performance.

[Comment 3.5]

Material and methods:

- Table 2. Indicate the significance level for the correlation analysis. Authors should indicate in the table which variables obtain a linear correlation.

[Response 3.5]

Thank you for your constructive comment. We have included the explanation of linear correlation in each variable and indicated the strength of the correlation as follows.

The correlation coefficients are listed in Table 2. HGS has a positive linear correlations with height and weight and a negative correlation with sex and percent body fat. Figure of 8 walk and TUG has a positive correlation with age, and the 30-s chair stand had a nega-tive correlation with age. Chair sit-and-reach shows a positive correlation with sex and a negative correlation with height. Since most of the FF variables show a weak linear rela-tion with the independent variables, it is possible to consider using nonlinear prediction models rather than linear prediction models to improve prediction accuracy.

[Comment 3.6]

Discussion:

I recommend extending the discussion with previous evidence of the relationship between the variables analysed.

[Response 3.6]

Thank you for your constructive comment. In the discussion section, we added the descriptions to discuss effect of correlation of input and output variable to the estimation accuracy using the results of the previous studies as follows:

The correlation coefficient shown in Table 2 represents the strength and direction of the linear relation between the input and output variables. In a previous study, height, weight, and BMI significantly correlated with HGS for older adults [25]. In this study, HGS had a higher correlation coefficient with these independent variables, and presented the most accurate estimation results compared to the other FF variables. From this results, we can infer that it is important to select input variables with strong correlation in order to ob-tain higher estimation results.

Round 2

Reviewer 1 Report

 The updated manuscript can be published.